# A Novel Toolbox for Automatic Design of Fractional Order PI Controllers Based on Automatic System Identification from Step Response Data †

Cristina I. Muresan [1], Iulia Bunescu [1], Isabela Birs [1,2,3,*] and Robin De Keyser [2,3]

1    Automation Department, Technical University of Cluj-Napoca, Str. Memorandumului No. 28, 400114 Cluj-Napoca, Romania
2    DySC Research Group on Dynamical Systems and Control, Ghent University, Technologiepark 125, B-9052 Ghent, Belgium
3    Flanders Make Consortium, EEDT Group, B-9052 Ghent, Belgium
*    Correspondence: isabela.birs@aut.utcluj.ro
†    This Paper Is an Extended Version of Our Paper Published in 16th International Conference on Dynamical Systems—Theory and Applications (DSTA), Online, 6–9 December 2021.

**Abstract:** This paper describes a novel automatic control toolbox, designed for non-experienced practitioners. Fractional order (FO) controllers are easily tuned with the main purpose of easy practical implementation. Experimental step data are required for the automatic FO controller tuning. An embedded system identification algorithm uses the step data to obtain a process model as a second order plus dead-time (SOPDT) system. Finally, the FO controller is computed based on the previously estimated SOPDT model in order to fulfil a set of user-imposed frequency domain performance specifications: phase margin, gain crossover frequency and gain margin maximization. Experimental step response data from a strongly nonlinear vertical take-off and landing unit have been used to design an FO controller using the toolbox. The experimental closed loop results validate the proposed toolbox. The end result is a user-friendly automatic fractional order controller tuning with endless possibilities of real-world applicability.

**Keywords:** control toolbox; automatic system identification; automatic tuning of fractional order controllers; experimental validation; vertical take-off and landing

**MSC:** 26A33; 37N35; 70Q05; 93C55

## 1. Introduction

Classical PIDs account for more than 90% of controllers used in the industry [1]. Even though several other control strategies have emerged, the popularity of the PID controller has been untouched. This is because PIDs are relatively simple and easy to use, even by the non-researcher. However, in the past two decades, the scientific community has experienced a fast-growing number of publications that discuss the topic of a generalization of the PID controller, namely the fractional order PID (FOPID) [2]. Most of the interest is focused on research and development [3,4], with very few practical applications in complex processes that the engineer would come across in the industrial sector [5]. This is partly due to the engineer's scepticism regarding fractional order controllers and to the researcher's failure of developing tuning and implementation methods that are simple enough to be used in practical applications.

However, FOPIDs greatly resemble the standard PID controller, much more than any other control strategy. The availability of a simple automatic tuning procedure for these controllers could pave the way towards the acceptance of such controllers as a viable option to replace the classical PID. Fortunately, FOPIDs come with extra advantages,

due to the supplementary tuning parameters and the fractional orders of integration and differentiation, which expand the flexibility of the classical PID controller [6,7].

Flexibility refers, first of all, to the iso-damping property of fractional order control systems [6–8]. The iso-damping property is closely related to an increased robustness and stability of control systems and can be achieved by ensuring a constant phase margin at the imposed gain crossover frequency. This can, in turn, be achieved by ensuring a flattened characteristic of the open loop phase at the gain crossover frequency. In this case, occasional gain variations do not lead to a phase change. This leads to stable and robust closed loop dynamic response despite modelling uncertainties [6,9–11]. Such a feature of the control system is highly desirable because, in industrial applications, modelling uncertainties and parameter variations frequently occur. A control system that exhibits the iso-damping property has the potential to minimize the impact of such modelling uncertainties.

Fractional order controllers achieve better control of time delay systems [5,12–14]. Most industrial plants exhibit time delays, so a controller that performs well in this case is preferred. The literature in this regard is quite extensive and covers various types of time delay systems, including multivariable ones [5,12–16].

One of the most important aspects of a control system consists of its ability to reject disturbances and avoid the negative impact these may have on the control performance. Fractional order controllers can achieve better disturbance rejection. They have supplementary tuning parameters, which can be used to address sensitivity for output disturbance, complementary sensitivity or reference to disturbance ratio for input disturbance [3,6,17–19].

Finally, fractional order controllers have the advantage of reducing the required control effort [5]. This aspect on its own represents a major advantage in industrial process control because of a more efficient energy use, combined with an additional increase in the life span of switching components. Even a slight energy reduction matters in industrial applications because this affects the quality, efficiency and sustainability of mass production systems. In manufacturing industries, all these small individual benefits add up to form a larger overall benefit [3]. Combining all of these advantages together implies better quality of the control system, which eventually triggers an improved production quality for the industry.

However, this flexibility comes with some important drawbacks. A major disadvantage of these fractional order PIDs consists in their actual implementation [20,21]. A digital form has to be obtained for the industrialization of such controllers. To achieve this, high-order conventional transfer functions are used to approximate the fractional order PID. These high-order transfer functions are generally complicated and require an increased number of computational resources. This leads to an increase in costs [20]. Several algorithms have been developed to approximate FOPIDs, but all of the well-known ones have one point in common: they are not easily handled by the non-researcher. A poor approximation is just as bad or even worse as a poorly tuned controller. Modern embedded software solutions nowadays can handle the additional implementation complexity easily. However, the engineer still requires an in-depth knowledge of fractional calculus and approximation methods to be able to produce a digital approximation of the fractional order controller. All of these issues have made it difficult to convey the advantages of fractional order controllers in industrial applications.

However, an automatic toolbox that does not require particular knowledge of fractional order control design and approximation could ease the transit of FOPIDs from research to industry.

The most widely used tuning methods for FOPIDs use a process model. However, in practice, accurate modelling of an industrial process usually means a model with increased complexity is necessary. Obtaining such a model for the process can become quite a time-consuming task. Fortunately, most of the time, for industrial processes, second order plus dead-time (SOPDT) models are sufficiently accurate and simple enough to be used in the tuning of a fractional order controller [22,23]. Process identification to determine the SOPDT model uses either open loop or closed loop responses [24–26], mostly in the form of a step response or relay produced one. Software-based approaches for identifying

SOPDT models are reviewed in [27]. Relay-based methods can be, however, disruptive in an industrial context, as they force the process to reach its stability limit and they could easily destabilize the process under the disturbance effect [28]. A safer method consists of the identification of a process model from step response data and then using analytical rules to produce the SOPDT model [29]. The most popular analytical rules are based on the use of graphics and areas [23], and they require system identification expertise to derive the model [29].

However, depending on the resources and the desired level of accuracy, manual labour for this task can be avoided and a significant amount of time can be saved. An alternative method, consisting of an automatic system identification algorithm, has been developed and described in [29]. The approach is fully automatic and does not require knowledge regarding system identification. Instead, the method requires a set of step response data as the only process information. The method is robust against disturbances, noise and system order [29], as has been demonstrated for various types of processes including non-minimum phase ones and over- and underdamped systems. In the current manuscript, an improved version of the algorithm in [29] is used. The automatic system identification routine returns a SOPDT model based on the process step response data:

$$G(s) = \frac{K}{(\tau_1 s + 1)(\tau_2 s + 1)} e^{-\tau_d s} \tag{1}$$

with $\tau_d$—the process dead-time, $K$—the process gain, $\tau_1$ and $\tau_2$—the time constants.

Once the model in (1) has been determined, this is used in the automatic design of a fractional order PI (FOPI) controller, described by the following transfer function:

$$C(s) = k_p \left( 1 + \frac{k_i}{s^\lambda} \right) \tag{2}$$

with the proportional and integral gains denoted as $k_p$ and $k_i$. The fractional order of integration is $\lambda \in (0, 2)$. Performance specifications regarding the gain crossover frequency, $\omega_c$, and phase margin, PM, are used to determine the $k_p$ and $k_i$ parameters. Analytical equations are derived for these two parameters as a function of $\lambda$, $\tau_1$, $\tau_2$, $\tau_d$ and $K$. The final controller parameters are selected as those that maximize the gain margin, GM, of the loop frequency response. At this point, the desired gain crossover frequency, $\omega_c$, and phase margin, PM, need to be specified by the user.

A discrete-time approximation of the previously designed FOPI controller in (2) is automatically produced next, along with the recurrent relation for the control signal. No input from the user is required at this step.

Several toolboxes for fractional order controllers and systems have been developed. The CRONE toolbox [30] is amongst the very first ones dealing with fractional calculus. It offers tools for fractional order modelling and system identification, as well as controller design. The design is not automatic, and the toolbox is not open-source. The user needs a fairly restrictive license, which has limited the use of the toolbox by academia and industry. Ninteger [31,32] is a toolbox that specifically targets approximations for fractional order systems. It does not provide any additional features for controller design or system identification. FOTF [33] addresses partially the same issues an Ninteger, allowing for the simulation of fractional order systems in general, including multivariable ones. It does not provide any algorithms for system identification on controller design. FOMCOM [34,35] is a toolbox similar to CRONE. It provides an identification module for the time and frequency domain identification of fractional order systems and a control module for FOPIF tuning. Approximation features are also included. A graphical user interface is available, which makes it more appealing to the user. However, FOMCOM, much like the other toolboxes requires knowledge of fractional order systems and control. FLOreS [36] is a newer toolbox designed for loop-shaping FOPIDs. It also comes with a graphical user interface and enables a frequency-domain design of both integer and fractional order controllers based

on integer or fractional order single-input–single-output processes. In this case, the process model has to be supplied by the user. In all cases, these toolboxes required multiple inputs and information from the user in order to design an adequate fractional order controller.

The novel toolbox presented in this manuscript incorporates all of the features presented: automatic SOPDT model identification using a set of step response data provided by the user; automatic FOPI controller design using only two performance specifications; automatic discrete-time approximation of the FOPI controller to produce a digital form ready to be implemented on dedicated devices, without requiring any input from the used. Thus, the toolbox we have developed requires a set of step response data and the values for the imposed frequency domain criteria: the gain crossover frequency and phase margin. The entire process identification, controller tuning and implementation is fully automatic.

The novelty of the manuscript resides in an improvement of the automatic system identification method in [29], followed by the automatic tuning and approximation of a FOPI controller to stabilize the process. Additionally, a novel user-friendly simple toolbox that does not require specific system identification nor control engineering knowledge is presented. The toolbox has been specifically designed for non-researchers in fractional order control and to minimize the amount of data required from the user. Developing such automatic tools that implement the generalized fractional order PID (FOPID) controller can facilitate their use in industrial systems.

The manuscript is divided into five sections. After the first introductory section, an overview of the automatic system identification approach is presented, along with the improvement that minimize the required data from the user. Section 3 presents the automatic design of the FOPI controller and its digital approximation. The novel toolbox and its utility are presented in Section 4, while the experimental results obtained using the toolbox on a vertical take-off and landing equipment are presented in Section 5. The concluding remarks are presented in Section 6.

## 2. Overview of the Novel Automatic System Identification Algorithm

The following input data need to be provided for the system identification algorithm:

1. A vector containing the process output from the step response $v(k)$, where $k = 0 \ldots N_s - 1$ and $N_s$ is the number of samples;
2. The sampling period $T_s$ that was used to acquire the process output data;
3. The step signal $u(k)$ supplied as the input to the process.

In the original system identification algorithm [29], an estimation of the process gain, $K$, has to be supplied. The algorithm has been improved to determine this automatically. The original output data collected from the process is first normalized to a unitary step response, $y(k)$, by dividing $v(k)$ by the amplitude of the input signal, $u(k)$. Then, the process gain, $K$, is computed. For a good approximation, the following formula is used:

$$K = \sum_{k=Ns-10}^{Ns-1} \frac{y(k)}{u(k)} \tag{3}$$

The next parameter to be approximated is the oscillation period, $T_p$. In the original approach [27], the $T_p$ value is estimated by the user. Here, the default value for $T_p$ is zero, such that it fits overdamped systems. In order to ensure that noise does not lead to a fake underdamped system, an incremental search is carried out in the proximity of the maximum value from $y(k)$. The error margin for noise rejection (EMNR) is 5%. For underdamped systems, in order to compute this parameter, the first two maximum amplitude points are determined. The first point is easy to determine as it will have the absolute maximum amplitude of the system. The second point is found by computing the maximum of the rest of the data.

Another computed parameter is $T_m$, the settling time. The error margin for noise rejection is 2%. The core idea for finding the required data is to find a sample of 15% of the data, where the amplitude lies in the range of 2% of the previously computed gain,

*K*. This also represents an improvement of the original approach [29], which adds to the automatization of the SOPDT model identification method.

The fourth parameter is $\tau_{dmax}$, which is the maximum time delay. In the original approach, the maximum time delay is also estimated by the user. Here, to estimate this value, the algorithm first checks for negative values in the output vector. Then, the algorithm searches for the index of the first point, which has the amplitude approximately 2% of the gain, *K*. That index is then multiplied by the sampling period, $T_s$, hence $\tau_{dmax}$ is obtained as integer order value of the sampling period. The minimum time delay value, $\tau_{dmin}$, is chosen to be always zero.

An additional feature of the automatic system identification routine implemented in the novel toolbox consists of the estimation of the process model. In order to estimate the order of the model, the algorithm computes both a SOPDT model and a first order plus dead time (FOPDT) one. Afterwards, the algorithm calculates the error between the model output, $m(k)$, for a unit step response and the process normalized output, $y(k)$, using the Mean Squared Error formula (MSE). The smallest of the two MSE values decides which is the better order for the model:

$$MSE = \frac{1}{N_s} \sum_{k=0}^{Ns-1}(y(k) - m(k)) \tag{4}$$

The estimation of the best time delay value is performed by considering small increments in the dead time, up to $\tau_{dmax}$. For each estimation of the dead-time, the algorithm replaces the measured step response, $y(k)$, by the delay free response, $s(k)$, which is later used to estimate the SO model parameters:

$$G(s) = \frac{K}{(\tau_1 s + 1)(\tau_2 s + 1)} = \frac{K\omega_n^2}{s^2 + 2\zeta\omega_n s + \omega_n^2} \tag{5}$$

where $\zeta$ is the damping factor and $\omega_n$ is the natural frequency. Once the model is estimated, the SOPDT in (1) is determined along with its step response, $\hat{y}(k)$. The Sum of Squared Errors (SSE) is computed:

$$SSE = \sum_{k=0}^{Ns-1}(y(k) - \hat{y}(k))^2 \tag{6}$$

The optimal value of the dead-time $\tau_d^*$ is selected as the one that minimizes (6):

$$\tau_d^* = arg\min_{\tau_d}SSE(\tau_d) \tag{7}$$

A detailed description of the estimation procedure is given in [29]. We briefly present hereafter some details regarding the parameter estimation for overdamped systems, as well as underdamped ones.

### 2.1. Estimation of a Second Order plus Dead Time Model from Overdamped Step Response Data

The measured step response of the SO model in (5) is used to calculate the area in continuous and discrete-time domains:

$$A = \int_0^{T_m}(K - y(t))dt \tag{8}$$

$$A = T_s \sum_{k=0}^{Ns-1}(K - y(k)) \tag{9}$$

Considering $x(t)$ as the step response of a first order system, $\frac{K}{\tau s + 1}$, with $\tau \triangleq \tau_1 + \tau_2$, then:

$$\tau = \frac{K}{A} \tag{10}$$

The step responses, $y(t)$ and $x(t)$, are given in Figure 1, where $S_1 + S_2 = 0$ and $A_1 = A_2 \triangleq A$ [28].

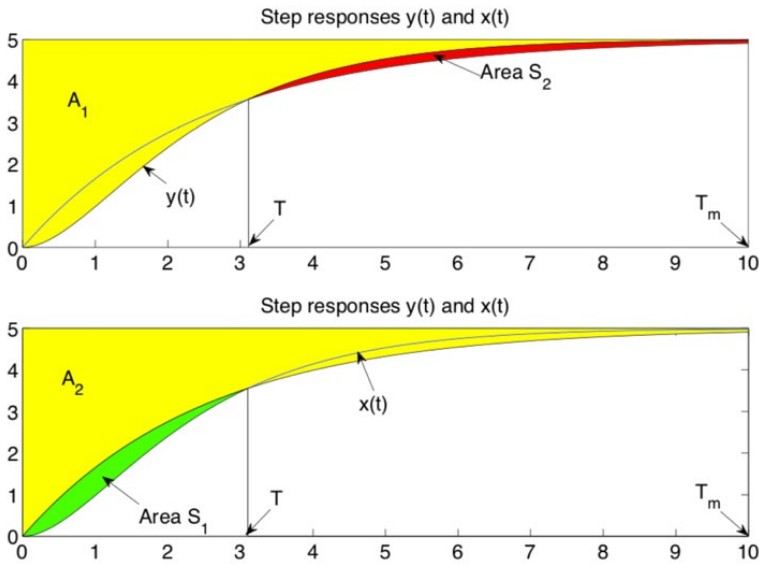

**Figure 1.** Corresponding areas for the algorithm on an overdamped system response.

The integral signal, $\Im(t)$, defined as the area between the previously defined $x(t)$ and the initial data $y(t)$, computed for $t \in [0, \ T_m]$, can be expressed in the discrete-time domain as:

$$\Im(k) = \Im(k-1) + T_s[x(k) - y(k)], \ \mathrm{k} = 0 \ldots N-1 \tag{11}$$

The integral signal is $\Im(0) = 0$ and $\Im(T_m) = 0$, leading to either a maximum or a minimum in the range $[0, \ T_m]$. An optimization routine is used next to determine:

$$T = \underset{t}{\mathrm{argmax}}|\Im(t)| \tag{12}$$

and the following area that corresponds to $T$ is defined as: $\mathrm{S} \triangleq \Im(T)$. The computation of the damping factor is then performed by solving $F(\xi) = 0$ as one of the following equations:

$$F(\xi) = \frac{S}{A} - e^{-\alpha}\left[1 - 2Re\left\{\frac{e^{-\alpha Z}}{1-Z^2}\right\}\right], \text{ if } 0.5 < \xi < 1 \tag{13}$$

$$F(\xi) = \frac{S}{A} - e^{-\alpha}\left[1 - \frac{e^{-\alpha Z}}{1-Z^2} - \frac{e^{-\alpha Z^{-1}}}{1-Z^{-2}}\right], \text{ if } 1 < \xi < 2 \tag{14}$$

where $Re\{\}$ is the real part of a complex number, $\alpha \triangleq \frac{T}{\tau}$ and $Z \triangleq \left(\xi - \sqrt{\xi^2 - 1}\right)^2$. Then, the natural frequency is $\omega_n = \frac{2\xi}{\tau}$.

This part of the algorithm is executed when the algorithm estimated $T_p = 0$.

### 2.2. Estimation of a Second Order plus Dead Time Model from Underdamped Step Response Data

For poorly damped processes, a typical step response as the one given in Figure 2 is produced. The idea behind this method is to rewrite the area A in (8) as:

$$\underline{A} = \int_0^{T_m} |K - y(t)|dt = A_1 - A_2 + A_3 - A_4 + \ldots \tag{15}$$

to account for the negative contributions of $A_2, \ A_4 \ldots$. Based on $y(t)$, the measured step response, the area in (15) is computed. The same area can be written analytically as:

$$\underline{A} = 2KRe\left\{\frac{\bar{p}/p}{\bar{p}-p}\left(-1 + 2\frac{e^{pT_1}}{1+e^{0.5pT_p}}\right)\right\} \tag{16}$$

with $p$ and $\overline{p}$ being the complex conjugated poles of a second order model (5), $\omega_n = \frac{2\pi}{T_p\sqrt{1-\xi^2}}$, $T_p$ is the period of oscillations and $T_1 = \frac{\arcsin(\xi)+\frac{\pi}{2}}{\omega_n\sqrt{1-\xi^2}}$. The algorithm then searches for the best damping factor, $\xi \in (0.01, 0.5)$, which makes:

$$\xi^* = \arg\min_{\xi}|F(\xi)| \tag{17}$$

where $F(\xi) = \left|\underline{A} - 2KRe\left\{\frac{\overline{p}/p}{\overline{p}-p}\left(-1 + 2\frac{e^{pT_1}}{1+e^{0.5pT_p}}\right)\right\}\right|$. Once the damping factor has been determined, the natural frequency is computed as $\omega_n^* = \frac{2\pi}{T_p\sqrt{1-\xi^{*2}}}$.

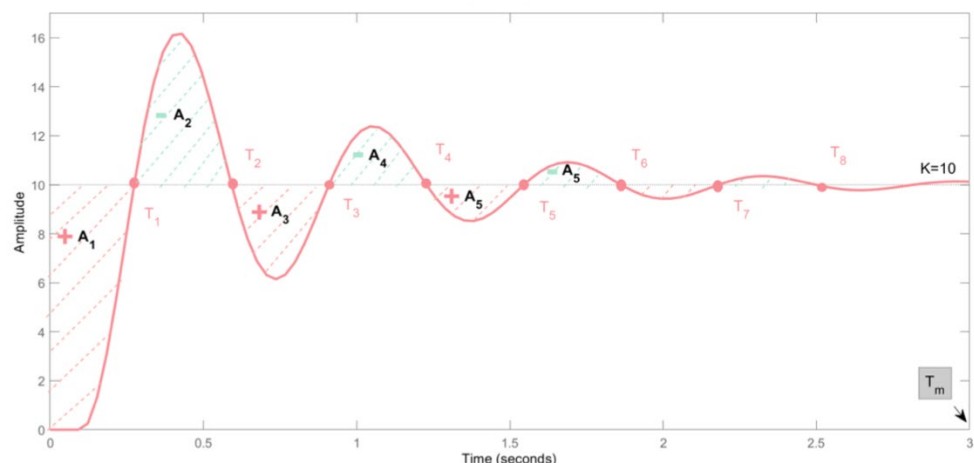

**Figure 2.** Corresponding areas for the algorithm on an underdamped system response.

This part of the algorithm is executed when the algorithm estimated $T_p > 0$.

## 3. Fractional Order Controller Design Method

For processes described as in (1), a FOPI controller is proposed, having the transfer function as:

$$C_{FO-PI}(s) = k_p\left(1 + \frac{k_i}{s^\lambda}\right) \tag{18}$$

where $0 < \lambda < 2$ is the fractional order and $k_p$ and $k_i$ are the proportional and integral gains. To determine these parameters, frequency domain performance criteria are imposed as follows [6,12,37]:

- A specific gain crossover frequency, $\omega_c$, leading to the magnitude equation:

$$|H_{ol}(j\omega_c)| = 1 \tag{19}$$

where $H_{ol}(s)$ stands for the transfer function of the open loop system and it is described as: $H_{ol}(s) = G(s) \cdot C_{FO-PI}(s)$.

- A phase margin, PM, leading to the phase equation:

$$\angle H_{ol}(j\omega_c) = -\pi + PM \tag{20}$$

Expanding (20), leads to:

$$-\tan^{-1}\left(\frac{k_i\omega_c^{-\lambda}\sin\frac{\lambda\pi}{2}}{1 + k_i\omega_c^{-\lambda}\cos\frac{\lambda\pi}{2}}\right) = X \tag{21}$$

where $X = -\pi + PM - \tan^{-1}\left(\frac{(p_1+p_2)\omega_c}{p_1 p_2 - \omega_c^2}\right) + \tau_d \omega_c$. Then, the parameter $k_i$ can be estimated using (21) as:

$$k_i(\lambda) = \frac{\tan(X)}{\omega_c^{-\lambda}\left(\sin\frac{\lambda\pi}{2} - \tan(X)\cos\frac{\lambda\pi}{2}\right)} \tag{22}$$

Based on (19), the $k_p$ can also be estimated as a function of the fractional order:

$$k_p(\lambda) = \frac{1}{|G(j\omega_c)|} \frac{1}{\sqrt{1 + 2k_i\omega_c^{-\lambda}\cos\frac{\lambda\pi}{2} + k_i^2\omega_c^{-2\lambda}}} \tag{23}$$

The automatic controller tuning algorithm computes the integral and proportional gains using (22) and (23) for each small increment in the fractional order, $0.01 < \lambda < 2$. At each step, the FO-PI controller transfer function is determined along with its continuous time approximation using the Oustaloup Recursive Approximation method [38]. The algorithm takes N = 6 as the approximation order, while the minimum and maximum frequencies for the approximation are selected depending on the gain crossover frequency, such as $\omega_{min} = 0.01\omega_c$ and $\omega_{max} = 100\omega_c$. The loop transfer function is then computed, and the corresponding gain margin, GM, is estimated. The algorithm searches for the best $\lambda$ value that maximizes the gain margin:

$$\lambda^* = \underset{\lambda}{\arg\max} GM \tag{24}$$

Once the optimal value, $\lambda^*$, has been determined, the corresponding $k_p^*$ and $k_i^*$ values are computed using (22) and (23). In order to avoid inappropriate results such as negative values being obtained for $k_i$ or $k_p$, the gain margin is not computed for such values and is attributed the value $-100$. The maximization of the GM allows for increased stability and robustness of the designed controller. In order to avoid unstable systems, if, after all iterations are performed, the value for the maximum gain margin is 0 or less, the introduced system is labelled as unstable and it points to the fact that either there is no possible solution for this type of implementation, or the input model inserted is incorrect.

To produce the discrete-time approximation of the FO-PI controller, the NRTF algorithm is used [20], which requires as inputs the sampling period, the maximum frequency for the discrete-time approximation, the order $N_d$ for the discrete-time approximation and the value for the parameter $0 \leq \alpha_d \leq 1$ in the discrete-time operator:

$$w_{NRTF}\left(z^{-1}\right) = \frac{1 + \alpha_d}{T_s} \frac{\left(1 - z^{-1}\right)}{1 + \alpha_d z^{-1}} \tag{25}$$

The sampling period, $T_s$, is automatically computed according to the Nyquist–Shannon sampling theorem, based on a search for the minimum value of the time constant in both the process (1) and the controller transfer function (18). The maximum frequency for the discrete-time approximation is then estimated as $\omega_{max} = \frac{\pi}{T_s}$. Finding the best values for $N_d$ and $\alpha_d$ is done using a *for loop* in which $N_d$ takes integer values from 2 and 8, and $\alpha_d$ varies with a step of 0.01 in the [0, 1] range. At each iteration of the loop, the MSE between the obtained discrete-time controller and the continuous one is computed. To avoid improper discrete-time controllers, a try-catch structure is used. After the loop has completed its execution, the minimum MSE is found, hence the corresponding discrete-time controller, $C_{dFO-PI}\left(z^{-1}\right)$. To implement the discrete-time controller on hardware pieces such as micro-controllers, PLC (Programmable Logic Controller), FPGAs (Field Programmable Gate Arrays), and process computers, a recurrence relation for the current control signal,

$c(k)$, is required as a function of both current and previous values of the error signal $\varepsilon(k)$, and on the previous control values. This is produced automatically using:

$$C_{dFO-PI}\left(z^{-1}\right) = \frac{a_0 + a_1 z^{-1} + a_2 z^{-2} + \cdots + a_{N_d} z^{-N_d}}{1 + b_1 z^{-1} + b_2 z^{-2} + \cdots + b_{N_d} z^{-N_d}} = \frac{c\left(z^{-1}\right)}{\varepsilon\left(z^{-1}\right)} \quad (26)$$

which leads to:

$$\begin{aligned} c(z^{-1}) + b_1 z^{-1} c(z^{-1}) + b_2 z^{-2} c(z^{-1}) + \cdots + b_{N_d} z^{-N_d} c(z^{-1}) = a_0 \varepsilon(z^{-1}) + \\ a_1 z^{-1} \varepsilon(z^{-1}) + a_2 z^{-2} \varepsilon(z^{-1}) + \cdots + a_{N_d} z^{-N_d} \varepsilon(z^{-1}) \end{aligned} \quad (27)$$

and to the final recurrence relation as:

$$\begin{aligned} c(k) = a_0 \varepsilon(k) + a_1 \varepsilon(k-1) + a_2 \varepsilon(k-2) + \cdots + a_{N_d} \varepsilon(k-N_d) - \\ -b_1 c(k-1) - b_2 c(k-2) - \cdots - b_{N_d} c(k-N_d) \end{aligned} \quad (28)$$

*Stability Analysis*

The characteristic equation of the closed loop system with the process defined as in (1) and the controller in (2) can be mathematically described using a quasi-polynomial [39]:

$$q(s) = Kk_p\left(1 + \frac{k_i}{s^\lambda}\right)e^{-\tau_d s} + (\tau_1 s + 1)(\tau_2 s + 1) \quad (29)$$

which is further rewritten as:

$$q(s) = \frac{K}{\tau_1 \tau_2}k_p s^\lambda + \frac{K}{\tau_1 \tau_2}k_p k_i + \left[s^2 + \frac{(\tau_1 + \tau_2)}{\tau_1 \tau_2}s + \frac{1}{\tau_1 \tau_2}\right]s^\lambda e^{\tau_d s} \quad (30)$$

The following notations are used hereafter: $b_0 = \frac{K}{\tau_1 \tau_2}$, $K_i = k_p k_i$, $a_1 = \frac{(\tau_1 + \tau_2)}{\tau_1 \tau_2}$, $a_0 = \frac{1}{\tau_1 \tau_2}$, $g = \tau_d s$ and $\lambda = \frac{a}{b}$. With these notations, (30) becomes:

$$q\left(\frac{g}{\tau_d}\right) = b_0 k_p \left(\frac{g}{\tau_d}\right)^{\frac{a}{b}} + b_0 K_i + \left[\left(\frac{g}{\tau_d}\right)^2 + a_1\left(\frac{g}{\tau_d}\right) + a_0\right]\left(\frac{g}{\tau_d}\right)^{\frac{a}{b}}e^g \quad (31)$$

The frequency response of (31) is defined as:

$$q(j\omega) = b_0 k_p \left(\frac{j\omega}{\tau_d}\right)^{\frac{a}{b}} + b_0 K_i + \left[\left(\frac{j\omega}{\tau_d}\right)^2 + a_1\left(\frac{j\omega}{\tau_d}\right) + a_0\right]\left(\frac{j\omega}{\tau_d}\right)^{\frac{a}{b}}(\cos\omega + j\sin\omega) \quad (32)$$

The frequency response in (32) can be written in terms of a real, $q_r(\omega)$, and an imaginary, $q_r(\omega)$, part as:

$$q(j\omega) = q_r(\omega) + jq_i(\omega) \quad (33)$$

Then, according to the Pontryagin and Hermite–Biehler theorems [40–42] and to ensure the interlacing property between the real and imaginary parts of (33), the range of values for $k_p$ and $K_i$ that ensure the closed loop stability are given by:

$$\max\left(-m_j(\omega)k_p - b_j(\omega)\right) < K_i < \min\left(-m_j(\omega)k_p - b_j(\omega)\right), \; j = 0, 1, 2 \ldots \quad (34)$$

and

$$k_p > -a_0 \quad (35)$$

where $m(\omega) = -\left|\Re(j)^{\frac{a}{b}}\right|\frac{\left|\omega^{\frac{a}{b}}\right|}{\left|L^{\frac{a}{b}}\right|}$ and $b(\omega) = b_1\left|\Re(j)^{\frac{a}{b}}\right|\frac{\left|\omega^{\frac{a}{b}}\right|}{\left|L^{\frac{a}{b}}\right|} - b_2\left|\Im(j)^{\frac{a}{b}}\right|\frac{\left|\omega^{\frac{a}{b}}\right|}{\left|L^{\frac{a}{b}}\right|}sign(\omega)$ with

$\Re$ and $\Im$ as the real and imaginary parts, $b_1 = \left[-a_0\cos\omega + \frac{\omega^2}{L^2}\cos\omega + \frac{a_1\omega}{L}\sin\omega\right]$ and

$b_2 = \left[ a_0 \sin \omega + \frac{\omega^2}{L^2} \sin \omega + \frac{a_1 \omega}{L} \cos \omega \right]$. The range for the integral gain can be determined using (34) and the previously defined notation $K_i = k_p k_i$.

## 4. A User-Friendly Toolbox

The toolbox AFOPI (Automatic FOPI) was developed as a Matlab Application and incorporates the entire automatic system identification procedure, followed by the FOPI design and discretization to finally return a recurrence relation as in (28). The toolbox is programmed to deliver gradual results throughout the procedure, hence it displays a result for each sub-process of the application. The interface of the program is presented in Figure 3, The application was designed using Matlab App Designer and can be added to any Matlab Apps library.

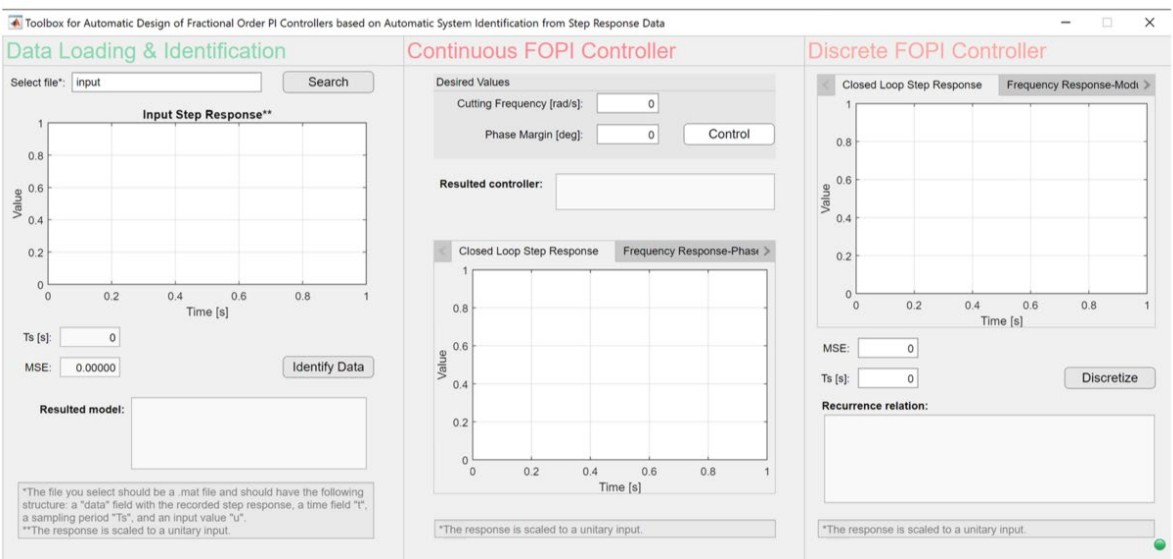

**Figure 3.** AFOPI toolbox.

The AFOPI application was split into three well-delimited sections. All the sections contain the title at the top and some disclaimer notes at the bottom. The leftmost section of the toolbox incorporates the identification process (Figure 4a). The input for this part will be inserted in subsection S1 and consists of a data structure with the following fields: $v(k)$—vector containing the measured step response of the process, *Ts*—sampling time of the recorded step, $u(k)$—the input of the recorded data. Subsection S2 provides the graphical output for the user, containing step response analysis for both the inserted and the identified system. The Identify Data button starts the automatic identification process. The last subsection, S3, displays numerical data to the user, namely the sampling period, *Ts*, the MSE between the inserted data and the model step response analysis, and the resulted model for the system.

The second section of the interface is presented in Figure 4b. In the first subsection, S1, the user is prompted to type the desired values for the phase margin and gain crossover frequency. The button Control triggers the automatic FOPI Controller design functions. The last part of this subsection displays the resulted Continuous FOPI Controller. The second subsection, S2, presents a carousel of graphs with numerous data analysis. The first is the closed loop step response analysis, the second contains the phase analysis of the open loop system via a Bode Diagram, the third the corresponding magnitude plot of this loop transfer function, and the fourth graph represents the control signal evolution, to a step input.

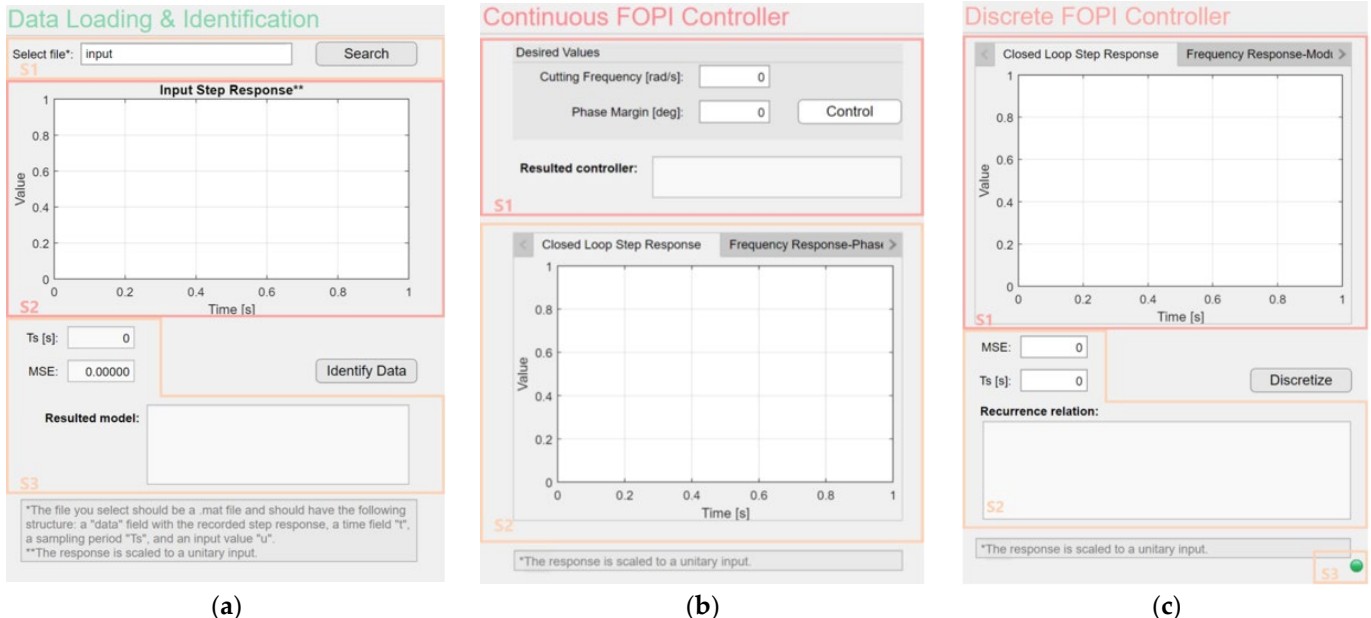

**Figure 4.** (**a**) Identification Section, (**b**) FOPI controller design Section, (**c**) Discrete-time FOPI controller Section.

The third and last section is found in Figure 4c. The first subsection, S1, of this part displays multiple graphical results which compare the performances of the continuous-time controller to the ones of the discrete-time controller. The data displayed is the step response of the closed loop systems and the frequency analysis of the open loop systems via a Bode Diagram. The button Discretize starts the computation of the discrete controller. The second subsection, S2, provides the user with the MSE value computed between the closed loop system step response with the continuous controller and the one with the discrete controller. It also shows the sampling period for the discretization. The last element of this subsection gives the user the final recurrence relation. The third subsection, S3, is active throughout all of the three sections and indicates when the application is busy doing computations by changing its colour to red.

## 5. Results and Discussion

The proposed method and the designed toolbox are validated using an experimental Vertical Take-Off and Landing (VTOL) unit [43], as indicated in Figure 5. The major component of the VTOL platform is the cantilever beam. This is connected to the base platform through a spinning rod. A variable speed fan is mounted on the right end of the cantilever beam, and a counterweight on its left. At one-third of the whole length of the beam, the anchoring point is located, near the counterweight. The input to the system is the voltage applied to the variable speed fan, which can take values between 0 V and 10 V. This allows the fan to spin and produce thrust, causing the beam to rise.

The output signal is the pitch angle, as indicated in Figure 5 and defined as the angle formed by two axes: one running thorough the centre of the beam and an imaginary vertical axis. When the two axes are perpendicular, the 0 position is considered. Movements in the interval $(-26°, 60°)$ are possible due to the physical structure of the platform. More in-depth details regarding the VTOL unit are presented in [43]. A NI Elvis board is used to supply input and collect output data from the VTOL unit.

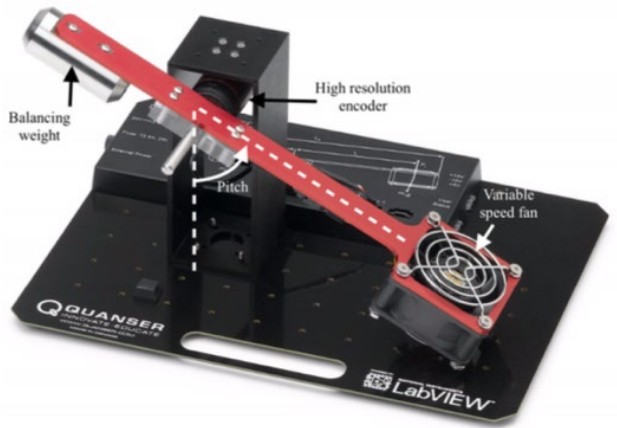

**Figure 5.** The case study: vertical take-off and landing platform.

The VTOL platform has been fed a 6.3 V step signal, leading to the experimental output signal used in (1). The sampling period of the input and output signals was $T_s$ = 0.005 s. The experimental output, as well as the model output are given in Figure 6. The automatic identification process achieved MSE = 0.02981, as indicated also in Figure 6, with the resulting process model given as:

$$G(s) = \frac{4.27}{0.1817s^2 + 0.1449s + 1} e^{-0.8s} \tag{36}$$

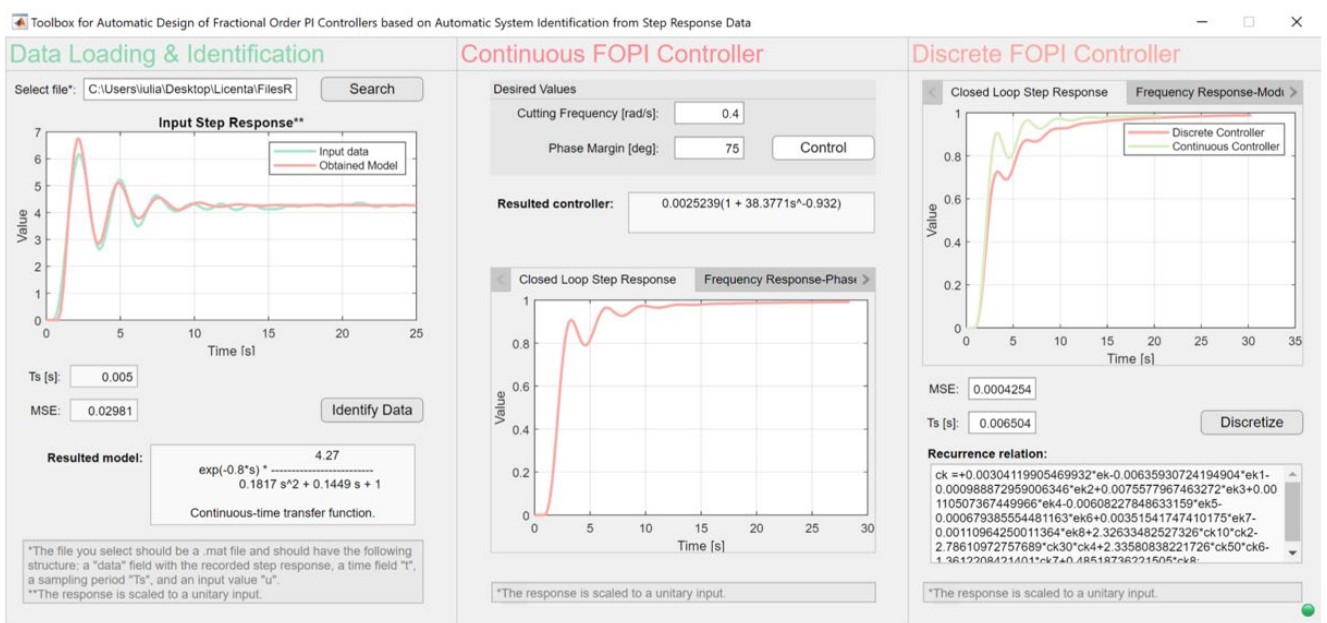

**Figure 6.** Result of the AFOPI toolbox for the case study.

To tune the controller, the design specifications were chosen as: $\omega_c$ = 0.4 rad/s and PM = 75°. Figure 7 shows the open loop gain margin, GM, as a function of the fractional order of integration. The maximum value of GM = 6.7009 dB is obtained for $\lambda = 0.932$. Then, using (22) and (23) the FOPI controller proportional and integral gains are computed based on (22) and (23), $k_p$ = 0.0025 and $k_i$ = 38.3771, with the final FOPI controller transfer function given as:

$$C_{FO-PI}(s) = 0.0025\left(1 + \frac{38.3771}{s^{0.932}}\right) \tag{37}$$

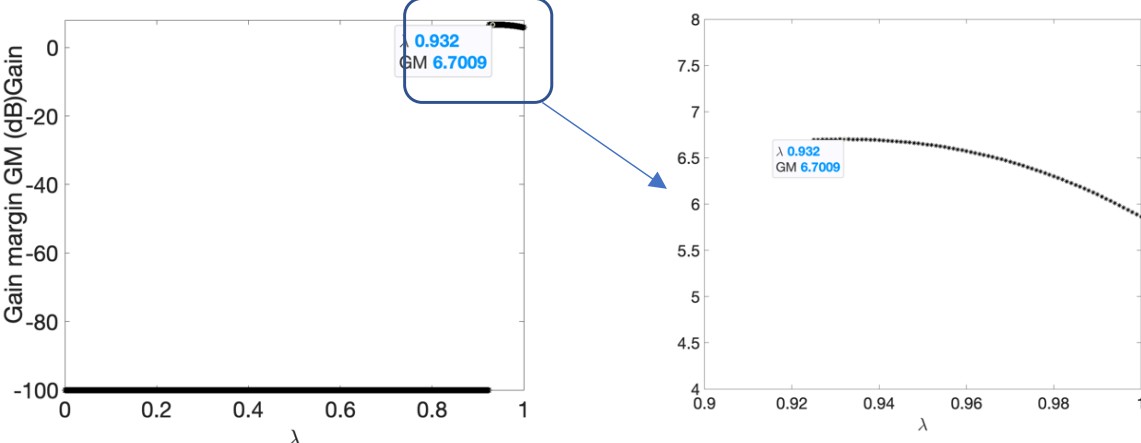

**Figure 7.** Gain margin of the open loop system.

The result of the automatic tuning algorithm is included in Figure 6, along with an estimated closed loop response. Figure 8 shows the frequency response of the resulting open loop system. Both performance criteria are met.

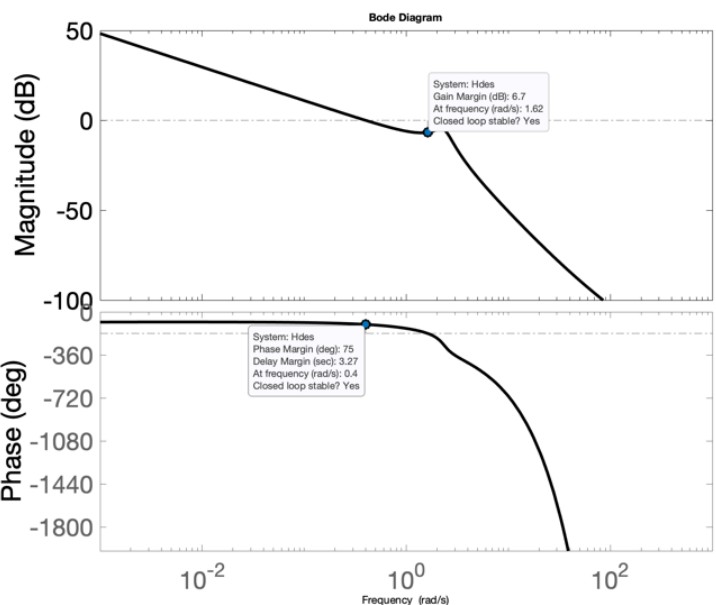

**Figure 8.** Bode diagram of the open loop system.

The discrete-time approximation of the designed FOPI controller was then obtained using the last section of the toolbox. The recurrence relation was further implemented on the VTOL unit. Figure 9 shows the closed loop experimental results. The best solution for the approximation of the discrete-time controller was obtained with an order 8, $\alpha_d = 0.4$ and a sampling period $T_s = 0.0065$ s, which resulted in the minimum MSE = $4.2510^{-4}$, as indicated in Figure 6. The closed loop step responses obtained using the continuous-time FOPI controller in (37) and the corresponding discrete-time approximation are also included in the last section of Figure 6. The recurrence relation was automatically computed, as also indicated in Figure 6. This was used to implement the discrete-time FOPI controller on the VTOL unit. The system is then given various references ranging from $-26°$ to $10°$. The experimental results are given in Figure 9a, for the pitch angle, and Figure 9b, for the voltage signal. The results show that the AFOPI toolbox can be used successfully to automatically design FOPI controllers that produce stable closed loop responses, even for poorly damped, nonlinear systems, such as the VTOL unit.

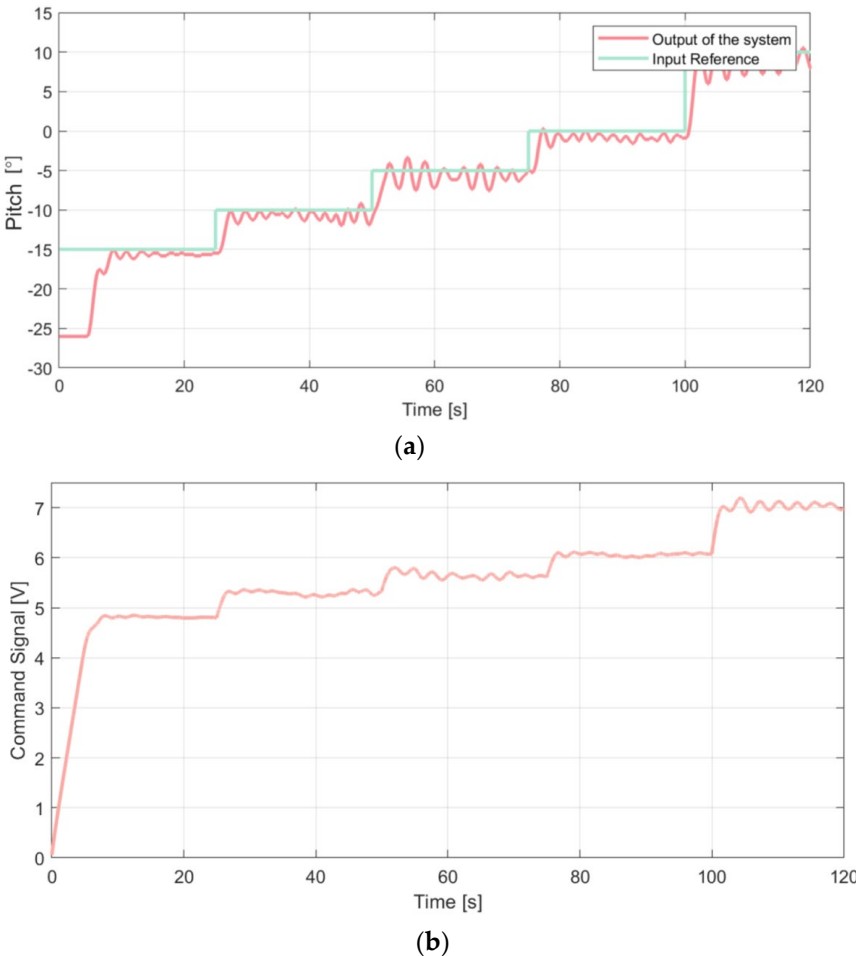

**Figure 9.** Closed loop experimental results for the VTOL unit using the FOPI controller designed using the proposed toolbox: (**a**) pitch angle, (**b**) voltage signal.

### 6. Concluding Remarks

A novel automatic fractional order control toolbox is presented based on a two-step algorithm: automatic system identification and automatic fractional order controller design and discretization. The toolbox is designed for non-experienced users. Very few input data from the user/engineer are required:

- the measured step response data acquired from the process to be controlled and the sampling period used to collect the output data for the automatic SOPDT model identification;
- the required performance criteria to design the FOPI controller: the loop phase margin and gain crossover frequency and phase margin, as measures of the final closed loop response.

The main benefit of the toolbox and the algorithms it uses consists of its simplicity. Such an automatic tool for designing FOPI controllers paves the way for other researchers to render fractional calculus more accessible to non-professionals and to increase the acceptance level of such controllers in the industrial sector. To exemplify the efficiency of the toolbox, experimental step response data from a nonlinear process have been used to determine first a SOPDT model and then a FOPI controller. The recurrent relation for the control signal, as generated automatically by the toolbox, is then implemented on an NI Elvis board. Experimental closed loop results obtained validate the AFOPI toolbox that can be used successfully to automatically design FOPI controllers that produce stable closed loop responses.

The toolbox, although versatile and easy to use, is limited to stable systems and to the tuning of FOPI controllers only. Although exemplified for single-input–single-output processes, the toolbox can be used to design FOPI controllers for multivariable systems in a decentralized approach. The user needs to decide according to a RGA analysis upon the input–output pairings. Then, for each input–output pair, the step input and corresponding output response are acquired by the user and then supplied as input data for the toolbox. Further research implies the design of FOPI controllers for multivariable systems, as well as experimental validation. Further research includes adding additional features in the toolbox to enable the design of fractional order PD and PID controllers.

**Author Contributions:** All authors contributed to the study conception and design. Material preparation, data collection and analysis were performed by I.B. (Iulia Bunescu), I.B. (Isabela Birs) and C.I.M. The first draft of the manuscript was written by C.I.M. and all authors commented on previous versions of the manuscript. All authors have read and agreed to the published version of the manuscript.

**Funding:** This work was partially supported by grants of the Ministry of Research, Innovation and Digitization, CNCS/CCCDI—UEFISCDI, project numbers PN-III-P1-1.1-TE-2019-0745 and PN-III-P1-1.1-PD-2021-0204, within PNCDI III, as well as CPNRR-III-C9-2022–I9, grant number 760018/27.01.2023. The authors have no relevant financial or non-financial interests to disclose.

**Data Availability Statement:** The experimental data generated during and/or analysed during the current study are available via an email request to the corresponding author.

**Conflicts of Interest:** The authors declare no conflict of interest.

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
