# Peer review of "A Novel Toolbox for Automatic Design of Fractional Order PI Controllers Based on Automatic System Identification from Step Response Dataâ€"

_mathematics, doi:10.3390/math11051097_

Round 1
Reviewer 1 Report
Some comments are given below:
1. Some recent results on PI control are suggested to be discussed in your introduction to highlight your contribution, such as Fuzzy tracking control for Markov jump systems with mismatched faults by iterative proportional-integral observers, Iterative proportional-integral interval estimation of linear discrete-time systems.
2. As it is known, PI control is suitable for scalar systems, therefore, my question is whether the proposed toolbox could be treated the multiple inputs multiple outputs.
3. In the title, I am attracted by the new, therefore, could you please give some comparisons with the existing results?

Reviewer 2 Report
This paper titled "A Novel Toolbox for Automatic Design of Fractional Order PI Controllers based on Automatic System Identification from Step Response Data" is well-written, and I suggest it for publication after minor revisions.
The authors have mentioned in detail the design of the FO-PI Controller; however, the proof of stability is not described. Please show the stability proof.
Reviewer 3 Report
The manuscript entitled “A Novel Toolbox for Automatic Design of Fractional Order PI 2 Controllers based on Automatic System Identification from 3 Step Response Data ” proposed control scheme is applied.
The manuscript is well-written and easy to understand, presenting a scientifically sound mathematical framework that is described in detail. The contributions of the paper are clearly articulated.
The experimental closed loop results validate the proposed toolbox. The end result is a user-friendly automatic fractional order controller tuning with endless possibilities of real-world applicability by QUARC.
The results obtained by simulations are clearly presented and discussed in an appropriate manner. However, before the paper can be considered for publication, there are some issues I would like the authors to address.
- In the Conclusion section, it is important to mention the limitations of the proposed research.
In the Conclusion section, please provide some insights into the limitations of the proposed research as well as potential challenges and directions for future research.
